# Neuroprotective Effects of Emodin against Ischemia/Reperfusion Injury through Activating ERK-1/2 Signaling Pathway

**DOI:** 10.3390/ijms21082899

**Published:** 2020-04-21

**Authors:** Stephen Wan Leung, Jing Huei Lai, John Chung-Che Wu, Yan-Rou Tsai, Yen-Hua Chen, Shuo-Jhen Kang, Yung-Hsiao Chiang, Cheng-Fu Chang, Kai-Yun Chen

**Affiliations:** 1Department of Radiation Oncology, Yuan’s General Hospital, 162 Cheng Kung 1st Road, Kaohsiung 80249, Taiwan; lwan@ms36.hinet.net; 2Core Laboratory of Neuroscience, Office of R&D, Taipei Medical University, 250 Wu-Hsing Street, Taipei 11031, Taiwan; m105095006@tmu.edu.tw (J.H.L.); ychiang@tmu.edu.tw (Y.-H.C.); 3Center for Neurotrauma and Neuroregeneration, Taipei Medical University, 250 Wu-Hsing Street, Taipei 11031, Taiwan; dr.jcwu@gmail.com (J.C.-C.W.); aggytsai@gmail.com (Y.-R.T.); swallows3366@gmail.com (Y.-H.C.); terbiun@gmail.com (S.-J.K.); 4Department of Neurosurgery, Taipei Medical University Hospital, 252 Wu-Hsing Street, Taipei 11031, Taiwan; 5Department of Surgery, School of Medicine, College of Medicine, Taipei Medical University, 250 Wu-Hsing Street, Taipei 11031, Taiwan; 6Department of Neurosurgery, Taipei City Hospital, Zhong Xiao Branch, 87 Tongde Rd., Nangang District, Taipei 11556, Taiwan; 7Graduate Institute of Neural Regenerative Medicine, College of Medical Science and Technology, Taipei Medical University, 250 Wu-Hsing Street, Taipei 11031, Taiwan

**Keywords:** ischemia/reperfusion, MCAO, Emodin, GLT-1

## Abstract

Background: Stroke is one of the leading causes of death and disability worldwide and places a heavy burden on the economy in our society. Current treatments, such as the use of thrombolytic agents, are often limited by a narrow therapeutic time window. However, the regeneration of the brain after damage is still active days, even weeks, after stroke occurs, which might provide a second window for treatment. Emodin, a traditional Chinese medicinal herb widely used to treat acute hepatitis, has been reported to possess antioxidative capabilities and protective effects against myocardial ischemia/reperfusion injury. However, the underlying mechanisms and neuroprotective functions of Emodin in a rat middle cerebral artery occlusion (MCAO) model of ischemic stroke remain unknown. This study investigates neuroprotective effects of Emodin in ischemia both in vitro and in vivo. Methods: PC12 cells were exposed to oxygen-glucose deprivation to simulate hypoxic injury, and the involved signaling pathways and results of Emodin treatment were evaluated. The therapeutic effects of Emodin in ischemia animals were further investigated. Results: Emodin reduced infarct volume and cell death following focal cerebral ischemia injury. Emodin treatment restored PC12 cell viability and reduced reactive oxygen species (ROS) production and glutamate release under conditions of ischemia/hypoxia. Emodin increased Bcl-2 and glutamate transporter-1 (GLT-l) expression but suppressed activated-caspase 3 levels through activating the extracellular signal-regulated kinase (ERK)-1/2 signaling pathway. Conclusion: Emodin induced Bcl-2 and GLT-1 expression to inhibit neuronal apoptosis and ROS generation while reducing glutamate toxicity via the ERK-1/2 signaling pathway. Furthermore, Emodin alleviated nerve cell injury following ischemia/reperfusion in a rat MCAO model. Emodin has neuroprotective effects against ischemia/reperfusion injury both in vitro and in vivo, which may be through activating the ERK-1/2 signaling pathway.

## 1. Introduction

Stroke is a common clinical disease with detrimental personal, social, and economic impacts [1]. Cerebral ischemia triggers a complex progression of biochemical and molecular mechanisms that impair neurologic functions in patients. Research has improved the treatment of stroke, yet there is no consensus regarding effective neuroprotective agents. Understanding the mechanism underlying the injury cascade triggered by ischemia is crucial for the discovery of neuroprotective agents. During the first few hours after ischemia, neurons in the ischemic penumbra or peri-infarct zone suffer transient reversible damage and may then ultimately undergo apoptotic cell death [2,3]. During ischemic injury, reactive oxygen species (ROS) and inflammatory reactions are important mediators of cerebral injury [4]. Evidence indicates that several signaling molecules and transcription factors are involved in ischemia-induced cell apoptosis [5,6]. These mediators include calcium/calmodulin-dependent kinases (CaMKs) and mitogen-activated protein kinases (MAPKs) such as extracellular signal-regulated kinase (ERK), p38, c-Jun N-terminal kinase (JNK), nuclear factor-κB (NF-κB), and the signal transducer and activator of transcription 1(STAT1) [7]. ROS activates MAPK cascades that contribute to cellular damage and induce the release of inflammatory cytokines, such as IL-1, IL-6, and tumor necrosis factor-α (TNF-α), activate caspase cascades, leading to cellular apoptosis [8]. Despite intensive investigations over recent decades, the molecular basis underlying the development and progression of ischemic injury is not yet completely understood.

Emodin is an anthraquinone derivative used in Chinese traditional medicine. Originally, it was used as a laxative, but it is also widely used to treat acute hepatitis [9]. In addition, Emodin possesses antineoplastic activity [10,11]. Recent research suggests a potential adjuvant role via antioxidative and other tumor regulatory effects in chemotherapy for human pancreatic cancer, chronic myelocytic leukemia, and hepatoma [9,12,13,14,15]. The neuroprotective effect of Emodin was first published in 2005 when its ability to interfere with the release of glutamate was identified as a method of neuroprotection [16]. Since then, both in vitro and in vivo experiments have shown the neuroprotective effects of Emodin against cerebral ischemia-reperfusion injury and glutamate-induced neural injury [17]. Although anti-growth effects may appear contradictory to its neuroprotective activity, various reports on the mechanisms of Emodin provide an explanation of its dichotomous role. The well-understood mechanisms of Emodin include its inhibition of different components of MAPK pathways and the inhibition of cell survival through inhibition of NF-κB family proteins. Moreover, its regulation of p38 MAPK, degradation of proteosomes, and inhibition of the PI3K-Akt pathway may also play a role in its neuroprotective activity [18,19].

Neuroprotection from ischemic neural injury by modulating glutamate transporter-1 (GLT-1) and glutamate levels is also interesting. The literature shows that glutamate has an excitatory effect that contributes to neuronal injury after ischemic injury [20]. On the other hand, GLT-1 expression decreases after ischemic neural injury and its suppression contributes to neuronal death [21,22]. Interestingly, intracortical delivery of GLT-1 via adeno-associated virus reduces ischemic damage in middle cerebral artery occlusion (MCAO) models [23]. The role of GLT-1 in neuroprotection should be explored further. In this report, we investigated the therapeutic effects of Emodin on ischemic brain injury and propose additional mechanisms explaining these neuroprotective effects involving glutamate and the glutamate transporter GLT-1.

## 2. Results

Cell viability under conditions of normoxia and oxygen-glucose deprivation (OGD)-hypoxia among the four time points is compared in Figure 1A. Initially, there was no difference between normoxia and OGD-hypoxia; however, cell viability decreased over time. Next, the toxicity of different doses of Emodin was evaluated (Figure 1B). When Emodin was applied to PC12 cells, cell viability significantly decreased after doses of more than 10 μM. Therefore, we applied OGD-hypoxia for 4 h and compared the results of Emodin at 0, 1, and 10 μM. The results are shown in Figure 1C. There were significant differences following Emodin treatment in PC12 cells after hypoxia for 4 h. In addition, a ROS assay was used to compare ROS generation in PC12 cells before/after hypoxia with/without Emodin (Figure 2A). ROS production in PC12 cells after hypoxia was greater than that of cells under nomoxia, but this was attenuated when hypoxic PC12 cells were treated with Emodin.

The expression of five proteins (ERK-1/2, GLT-1, Bcl-2, active caspase-3, and β-actin) were studied in four experimental groups, normoxia with/without Emodin and hypoxia with/without Emodin. Protein expression levels and the ratio of protein expression relative to that of the normoxia group are shown in Figure 2B,C. The relative ratio to the normoxia group was less than 1 in the hypoxia group for all proteins except active caspase-3. After Emodin treatment, the ratios of phosphorylation ERK-1/2 and GLT-1 expression increased for both the normoxia and hypoxia group. For the hypoxia group, expression levels of ERK-1/2, GLT-1, and Bcl-2 increased following Emodin treatment. For active caspase-3 expression, the relative ratio decreased to 1 in the Emodin-treated hypoxia group.

In addition, glutamate levels, which were determined from cell culture medium collected after 24 h normoxia or OGD-hypoxia with three different Emodin doses, are shown in Figure 2D. There were no significant differences among the different doses for normoxia; however, a significant difference was observed between the two hypoxia groups (with/without Emodin).

Rats pretreated with Emodin exhibited smaller infarct sizes (white area) after MCAO, and the total volume of the infarcts was significantly different (Figure 3A,B). The difference in volume was at least 50 mm^3^ between rats pretreated with or without Emodin. Moreover, we observed a significant decrease in both Terminal Deoxynucleotidyl Transferase dUTP Nick End Labeling (TUNEL) staining and NeuN-immunoreactivity in the Emodin-treated rats (Figure 3C). In addition, we found that rats pretreated with Emodin exhibited a significantly higher percentage of recovery in body asymmetry than nontreated rats (Figure 3D).

GLT-1 is responsible for the vast majority of functional uptake of extracellular glutamate in the central nervous system. Thus, GLT-1 expression was examined by immunohistochemical staining of rat brain slices (Figure 4A). GLT1 protein expression was higher in Emodin-treated rats than in nontreated rats (Figure 4B). In addition, expressions of p-ERK and Bcl2 increased in Emodin-treated rats while expression of active caspase-3 decreased expectantly (Figure 4B,C).

## 3. Discussion

Recent research into ischemic brain injury has revealed the importance of GLT-1 for neuroprotection. Glutamate has an excitatory effect that contributes to neuronal injury after an ischemic insult, while GLT-1 expression decreases after ischemic injury and its suppression contributes to neuronal death [21,22]. Interestingly, intracortical delivery of GLT-1 via an adeno-associated virus reduces ischemic damage in MCAO models.

GLT-1 expression following Emodin-mediated neuroprotection was also associated with decreased extracellular glutamate levels. Glutamate levels increase after ischemia/reperfusion injury and mediate neurotoxicity after stroke. Our experimental results suggest that Emodin treatment correlated with increased GLT-1 expression and decreased extracellular glutamate levels.

In addition to decreasing extracellular glutamate levels, Emodin treatment was also associated with decreased ROS generation. Since ROS contributes to ischemia/reperfusion injury in myocardial and cerebral infarction, decreased ROS levels should reduce the severity of injury and lead to improved outcomes. Previous studies have used glutamate to induce ROS and demonstrated mitigation of ischemia/reperfusion injury through the use of Emodin [17]. Our investigation used a rat MCAO model [24] to demonstrate in vivo neuroprotective effects of Emodin, and the results concur with a previous in vivo study using a focal ischemia monofilament occlusion model [17]. Other effects of Emodin include antiapoptotic effects following ischemia/reperfusion injury through increasing the expression of Bcl-2 and decreasing the expression of active capsase-3. A previous study suggested that GLT-1 may be regulated by the ERK-1/2 signaling pathway [25], as Emodin increases the expression of GLT-1 through activation of ERK-1/2.

Emodin is a promising chemopreventive and chemotherapeutic agent for brain injury [17,26,27]. Studies have shown that Emodin has anti-inflammatory effects by modulating the immune system in various inflammatory disorders including Alzheimer’s disease, pancreatitis, arthritis, asthma, atherosclerosis, myocarditis, and glomerulonephritis [28].

As an anti-inflammatory agent, Emodin can ameliorate lipopolysaccharide-induced microglial activation and apoptosis [29] and can reduce pro-inflammatory cytokine and chemokine expression in human umbilical vein endothelial cells (HUVECs) [30]. Moreover, Emodin also improves myocardial ischemia/reperfusion injury via suppression of pro-inflammatory cytokines (TNF-α and NF-κB) and apoptosis (caspase-3) [31]. Furthermore, Emodin can inhibit Aβ-induced neurotoxicity [32] and can ameliorate cycloheximide-induced impairment of memory consolidation in rats.

Recently, Guang et al. revealed that Emodin increases atrial natriuretic peptide (secretion via activation of K^+^ATP channels in cardiac atria [33]. In this study, we demonstrated that the protective effects of Emodin on ischemia injury included antiapoptotic and anti-ROS effects. Emodin increased the expression of phosphorylated ERK-1/2 and Bcl-2 in OGD-induced cell injury. In addition, Emodin decreased ROS generation and caspase-3 expression.

Despite the many functions, the mechanisms of Emodin in stroke are still under investigation and discussion. The pathways underlying Emodin-induced neuroprotection are not yet conclusive. In our investigation, decreased glutamate levels were observed after MCAO. In addition, the area of ischemic injury was reduced with Emodin treatment. Furthermore, increased numbers of NeuN- and TUNEL-expressing cells were observed in Emodin-treated rats. The decrease in ischemic injury and the increase in neural cells demonstrated the protective effects of Emodin.

Other mechanisms explaining the neuroprotective effects of Emodin have been proposed. One of the major pathways considered is the activin A pathway [26]. Emodin-mediated inhibition of inducible nitric oxide synthase also demonstrates its protective effects in alleviating brain injury after blast-induced traumatic brain injury [34]. Since Emodin exhibits favorable neuroprotective activity, its potential toxicity and feasibility for clinical use is of concern for future applications. In our study, Emodin exhibited mild cytotoxicity. Cells cultured under high concentrations of Emodin (Figure 1B) demonstrated decreased cell viability. Although mild cytotoxicity was observed, it is also noteworthy that previous clinical trials involving Emodin used doses of up to 60 mg/kg in humans [35]. For future clinical trials, a safe and effective therapy should be achievable.

There are several limitations to our study. First, we found that the effects of Emodin were associated with upregulation of glutamate transporters. GLT-1 expression increased after Emodin treatment under hypoxic conditions both in vitro and in vivo. However the detailed mechanisms of GLT-1 regulation by Emodin are unknown and remain to be explored in future. Second, our data showed that GLT-1 is upregulated in PC12 cells and cortex after Emodin treatment. It is reported that GLT-1 is the major glutamate transporter and is primarily expressed in astrocytes as well as in neuron and in axon terminals; whether Emodin induces GLT-1 in astrocytes is unclear and needs to be examined.

Other agents associated with upregulation of GLT-1 have been documented previously. One potential candidate associated with GLT-1 is ceftriaxone [36]. Interestingly, despite ceftriaxone-mediated increases in GLT-1 activity, the PASS (Preventive Antibiotic in Stroke Study) clinical trial demonstrated insignificant effects in patients receiving ceftriaxone. [37] Although ceftriaxone increased GLT-1 activity, it did not lead to a significant improvement in clinical outcomes for stroke patients; thus, it would be interesting to investigate whether Emodin’s ability to increase GLT-1 expression and the resultant decrease in extracellular glutamate could lead to significant neuroprotection in stroke patients.

Additional investigations and clinical trials of Emodin for the treatment of stroke are needed. The significance of other mechanisms involved in Emodin-mediated neuroprotection also deserve attention. Inhibition of GLT-1 after Emodin treatment should provide further information on the role of the multiple mechanisms that underlie the protective effects of Emodin. Furthermore, an increased understanding of the mechanisms underlying the neuroprotective effects of Emodin will be beneficial for the design of clinical trials as well as inclusion and exclusion criteria for patients in these trials. Lastly, while the currently known mechanisms of Emodin appear scientifically sound and the safety issues of Emodin use appear manageable, carefully designed clinical trials will be crucial to determine its significance as a neuroprotective agent in stroke.

## 4. Material and Methods

### 4.1. MCAO

Animals were anesthetized before the MCAO surgery according to a Taipei Medical University Laboratory Animal protocol. Animal studies were approved by the Institutional Animal Care and Use Committee (IACUC) of the Taipei Medical University. Adult male Sprague–Dawley rats (10–12 weeks, 250–300 g) were used in this study and anesthetized with Zoletil (50 mg/kg, Vibac, France) mixed with xylazine (10 mg/kg, Rompun, Bayer, Germany). Animals were divided into two groups: vehicle (Saline) and Emodin (15 mg/kg, i.p.), with *n* = 12 in each group. The right middle cerebral artery (MCA) is ligated with a 10-O suture using methods previously described. [24] The ligature is removed after 60-min ischemia to generate reperfusion injury. Core body temperature will be monitored with a thermistor probe and maintained at 37 °C with a heating pad during anesthesia. After recovery from the anesthesia, body temperature is maintained at 37 °C using a temperature-controlled incubator. Using this animal model, our group has generated consistent ischemic damage in rodent brains, demonstrated by brain infarction visualization and behavioral analysis.

### 4.2. Brain Tissue Staining

One day after MCAO surgery, animals were anesthetized and perfused intracardially with saline. Rat brains were sectioned into 2-mm-thick slices and immediately incubated in a 2% 2,3,5-triphenyl-tetrazolium chloride (TTC) solution (Sigma-Aldrich, St. Louis, MO, USA) for 15 min at room temperature and then transferred into a 4% paraformaldehyde solution for fixation. Damaged brain tissue loses the ability to metabolize TTC into a red color. The infarct volume of the lesioned side was measured together with the sum of the TTC-unstained region.

### 4.3. Immunohistochemical Staining and Terminal Deoxynucleotidyl Transferase dUTP Nick End Labeling (TUNEL) Staining

Animals were anesthetized and perfused transcardially with saline followed by 4% paraformaldehyde (PFA) in phosphate-buffered saline (PBS; 0.1 M, pH 7.2). Rat brains were serially transferred into 20% and 30% sucrose in PBS overnight. Brains were sectioned into slices of 10–12 mm thickness on a cryostat and stored at −30 °C. Brain slices were immersed in PBS for 30 min and then incubated in PBS containing 0.2% Triton X-100 (PBST) for 15 min at room temperature. Brain sections were incubated with 3% PBS after removing the endogenous peroxidase activity with 3% H_2_O_2_ in PBS for 10 min and incubated overnight at 4 °C with 10% normal goat serum-PBST. After blocking in goat serum-PBST, the brain sections were incubated with the primary antibody diluted 1:100 with 10% normal goat serum-PBST. Then, sections were rinsed with PBS three times for 5 min each time and incubated with a secondary antibody linked with polymer-horseradish peroxidase (HRP) for 1 h. After incubation with the secondary antibody and washing in PBST (3 × 5 min), sections were incubated for 10 min with 0.04 mg of 3,3′-diaminobenzidine (DAB, in 200 mL distilled water) (DAKO Corporation, Hamburg, Germany). The DAB-stained sections were rinsed with PBS (3 × 10 min) to halt the chromogen reaction, wet-mounted onto gelatin/chromium-coated slides, and allowed to air-dry overnight. TUNEL staining was performed using an In Situ Cell Death Detection Kit (Roche, Philadelphia, PA, USA), according to the manufacturer’s protocol.

### 4.4. Body Asymmetry

Behavioral measurements were evaluated by examining body asymmetry. Body asymmetry was quantitatively calculated with the lifted body swing test [24]. Rats were examined for lateral turning when raising their tails above the experimental table. The number of first turns of the head or upper body contralateral to the ischemic side was counted in 20 consecutive trials and was normalized as follows: % recovery = [1 − (lateral turns in 20 trials − 10)/10] × 100.

### 4.5. Cell Culture and Oxygen-Glucose Deprivation (OGD)-Hypoxia

PC-12 cells were purchased from the American Type Culture Collection (CRL-1721). Cells plated on precoated poly-D-lysine dishes were incubated in Roswell Park Memorial Institute (RPMI)-1640 medium including 10% heat-inactivated horse serum (HS) and 5% fetal bovine serum (FBS) and maintained in a 5% CO_2_ incubator at 37 °C. OGD culture to mimic ischemia-reperfusion injury was performed as previously described [38,39]. Cell medium was exchanged with glucose-free RPMI-1640 medium including 2% HS and 1% FBS, and then, cells were transferred into an aerobic incubator with a continuous gas flux (95% N_2_/5% CO_2_) for 4, 8, and 24 h at 37 °C. Then, cell media was replaced with normal media and cells were reoxygenated in a 5% CO_2_ incubator at 37 °C.

### 4.6. Viability Assay

Cells were seeded in 24-well plates (1 × 10^5^ cells per well) for 24 h. Cells were then exposed to OGD or treated with different doses of Emodin for 24 h, after which the cells were incubated with 5 mg/mL 3-(4,5-dimethylthiazol-2-yl)-2,5-diphenyltetrazolium bromide (MTT) solution for 4 h. Cells metabolized MTT into formazan that was dissolved by the addition of DMSO (200 μL/well). Cell viability was measured by analyzing the absorbance at 490 nm for each well with an ELISA 96-well plate reader. Results are expressed as the percentage of viable cells detected following OGD compared to that of control normoxic plates.

### 4.7. ROS Production

Cytosolic ROS production was measured using 2′,7′-dichlorofluorescien diacetate (DCFDA) which is converted into fluorescent 2′,7′-dichlorofluorescien (DCF) by oxidation. Cells were seeded in 6-well plates (1 × 10^6^ cells per well) and allowed to adhere overnight. After cell incubation under normoxia or OGD conditions in the presence or absence of Emodin for 4 h, cells were incubated with DCFDA for 30 min at 37 °C and washed with PBS. DCF fluorescence was determined using a multi-well fluorescence spectrophotometer (Varioskan Flash, Thermo Scientific, Waltham, MA, USA) with 490-nm excitation and 525-nm emission filters.

### 4.8. Western Blotting

Cells and rat frontal cortex (FC) tissue were lysed by the addition of RIPA buffer with a protease inhibitor cocktail. Proteins (50 μg) from cell lysates were diluted in 5× SDS buffer and denatured at 95 °C for 5 min. Proteins were electrophoresed on 10% or 12% SDS-polyacrylamide gels and electro-transferred onto a polyvinyldifluoride membrane. After incubation for 1 h in blocking solution (5% nonfat dry milk with TBST), membranes were incubated with primary antibodies overnight. After washing 3 times in TBS-Tween-20, membranes were incubated with goat HRP-linked anti-mouse and anti-rabbit IgG antibodies for 1 h and then developed with an enhanced chemiluminescence plus detection kit (Amersham Life Sciences, Piscataway, NJ, USA). All results were normalized to the levels of β-actin, which was used as a loading control, and the amount of immunoreactivity was expressed relative to the corresponding control.

### 4.9. Glutamate Release Assay

Cells were seeded in 6-well plates (1 × 10^6^ cells per well) and allowed to adhere overnight. The next day, cell medium was exchanged with glucose-free RPMI-1640 medium including 2% HS and 1% FBS, and then, cells were transferred into an aerobic incubator with a continuous gas flux (95% N_2_/5% CO_2_) for 4 h at 37 °C. Then, extracellular glutamate was collected and measured with a glutamate assay kit (Catalog No. MAK-330, Sigma-Aldrich, St. Louis, MO, USA) according to the manufacturer’s instructions. Glutamate levels were measured using an F-4500 fluorescence spectrophotometer (Hitachi, Tokyo, Japan) and normalized to the protein concentration. The standard curve for the glutamate level is 0–2.5 mM. The coefficient of variation is 12.5%.

### 4.10. Statistical Analysis

Statistical analyses were performed with a Student’s *t*-test and one-way or two-way ANOVA, with Newman–Keuls post hoc test for statistical comparison. Significance was inferred at *p* < 0.05. Data are presented as the means ± standard deviations (SEMs).

## 5. Conclusions

In conclusion, our investigation supports previous findings that Emodin serves as a neuroprotective agent in stroke. In addition, decreased ROS production, increased GLT-1 expression, and decreased extracellular glutamate levels are associated with Emodin treatment in a rat MCAO model (Figure 5). Further confirmation of other possible mechanisms involved should be helpful for understanding the neuroprotective characteristics of Emodin.

## Figures and Tables

**Figure 1 ijms-21-02899-f001:**
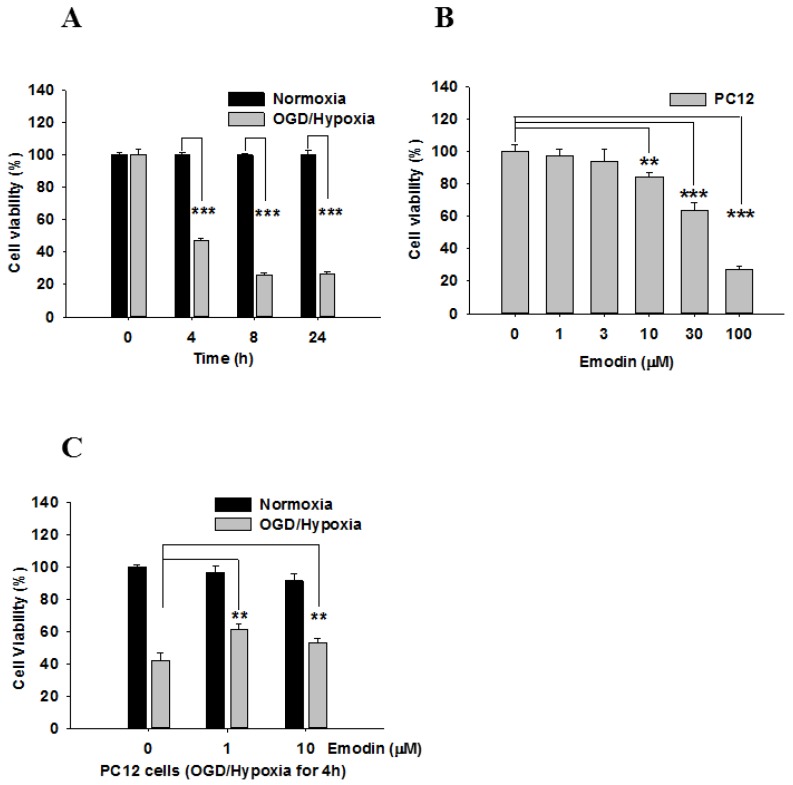
Emodin suppresses OGD/hypoxia-induced cell death in PC12 cells: Cell viability was determined by 3-(4,5-dimethylthiazol-2-yl)-2,5-diphenyltetrazolium bromide (MTT) assay. (**A**) Comparison of cell viability between normoxia and OGD-hypoxia at different time points. (**B**) Cell viability with different doses of Emodin. (**C**) The protective effect of Emodin under OGD/hypoxia. Data represent SEMs of three independent experiments. ** *p* < 0.01, *** *p* < 0.001 using Student’s *t*-test.

**Figure 2 ijms-21-02899-f002:**
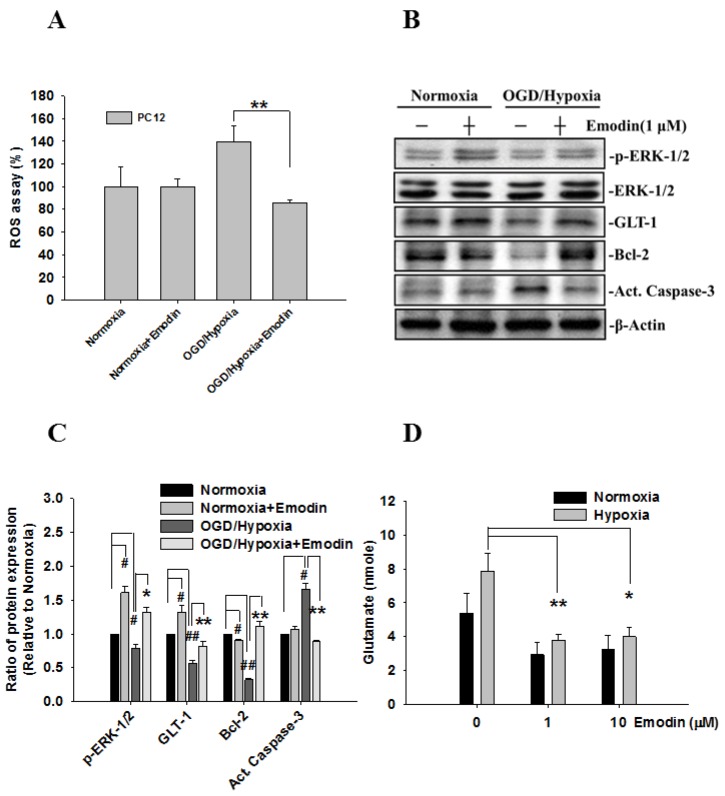
The effects of Emodin treatment of PC12 cells after OGD/Hypoxia on reactive oxygen species (ROS) production, glutamate release, and apoptosis are regulated through the extracellular signal-regulated kinase (ERK)-1/2 signaling pathway. (**A**) The protective effects of Emodin on ROS under OGD-hypoxia. (**B**) Phosphorylation levels of ERK-1/2 and the expression of GLT-1, Bcl-2, and activated caspase-3 (Act. Caspase-3) were examined by western blotting. (**C**) Bar graph showing semi-quantified densitometry. The ratio of p-ERK-1/2 was normalized to the levels of ERK-1/2 and β-Actin. (**D**) Glutamate was collected from cell culture medium after normoxia or OGD-hypoxia for 24 h and was detected with a colorimetric assay. The # and * indicate separate comparisons with normoxia and OGD-hypoxia conditions. * *p*, # *p* < 0.05 and ** *p*, ## *p* < 0.01 using Student’s *t*-test.

**Figure 3 ijms-21-02899-f003:**
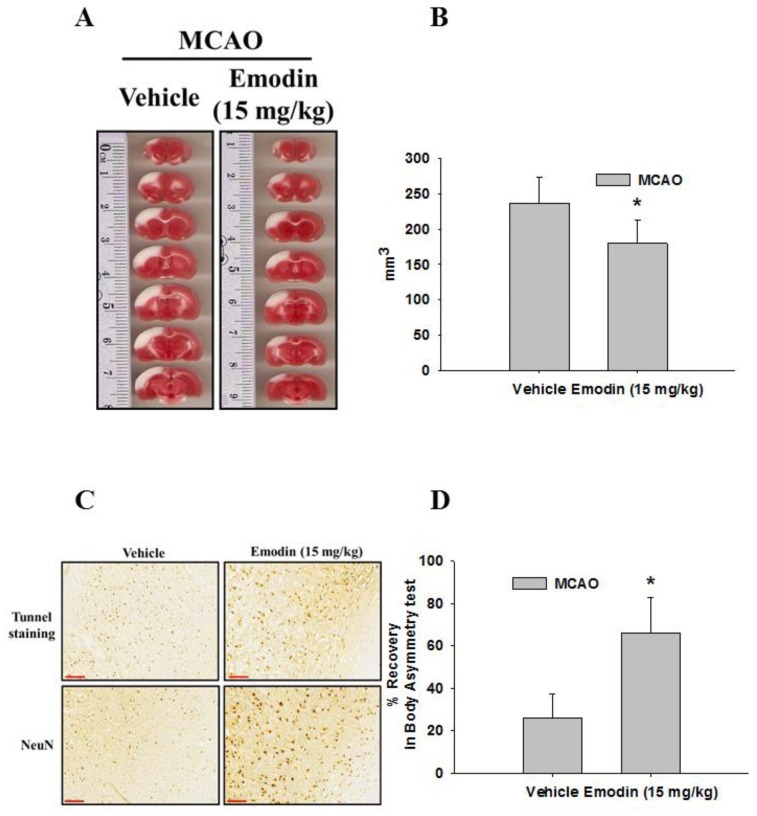
Emodin attenuates infarction sizes after middle cerebral artery occlusion (MCAO) in rats and improves the recovery in body asymmetry. (**A**) Brain sections were stained with 2,3,5-triphenyl-tetrazolium chloride (TTC) and (**B**) total volume of the infarction (the vehicle and emodin groups, *n* = 6 in each group. (**C**) Terminal Deoxynucleotidyl Transferase dUTP Nick End Labeling (TUNEL) and NeuN staining were observed in brain slices following ischemia/reperfusion injury. Scale bar = 100 μm (**D**) Body asymmetry was tested after MCAO surgery. Data represent SEMs of three independent experiments. * *p* < 0.05 using Student’s *t*-test.

**Figure 4 ijms-21-02899-f004:**
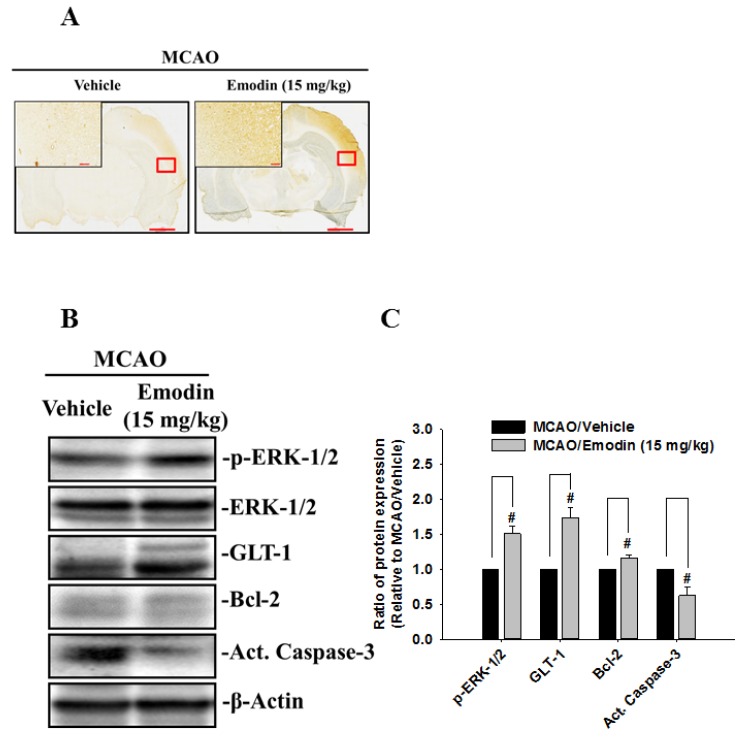
Emodin increases GLT-1 protein expression in rat brain sections after MCAO. (**A**) GLT-1 was detected by immunohistochemical staining of rat brain slices after MCAO surgery. Red box denotes area shown in higher magnification inset on the upper left. Scale bar of low-power photomicrographs is 2 mm. Scale bar of high-power photomicrographs (higher left corner) is 100 μm. (**B**) Expression of p-ERK-1/2, GLT-1, Bcl-2, and Act. Caspase-3 were determined by western blotting of rat brain cortex. (**C**) Bar graph showing semi-quantified densitometry. The ratio of p-ERK-1/2 was normalized to the levels of ERK-1/2 and β-Actin. # indicates comparison with MCAO/vehicle and MCAO/Emodin. # *p* < 0.05 using Student’s *t*-test. (*n* = 3).

**Figure 5 ijms-21-02899-f005:**
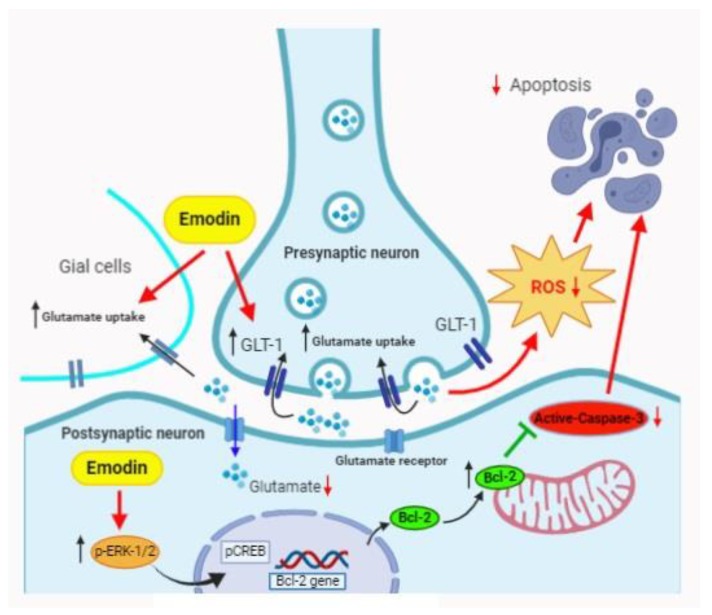
The proposed mechanisms of Emodin on the protection effect in a rat MCAO model: Emodin prevents glutamate release through GLT-1 and decreases ROS production, resulting in suppression of apoptosis. Additionally, Emodin stimulates p-ERK1/2 expression and blocks caspase-3 to decrease apoptosis via the regulation of Bcl-2 gene.

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
