# Peer review of "Neuroprotective Effects of Emodin against Ischemia/Reperfusion Injury through Activating ERK-1/2 Signaling Pathway"

_ijms, 2020, doi:10.3390/ijms21082899_

Round 1
Reviewer 1 Report
This is a very interesting article describing the results of a properly designed experiment.
I have a few comments:
1. Title - I suggest deleting in vitro and in vivo.
2. Introduction:
I suggest adding literature to the first sentence e.g. Ann Clin Biochem. 2017;54(3):378-385.
STAT1 - please provide full name (signal transducer and activator of transcription 1).
Give a name "Emodin" with a lowercase.
3. Results:
the quality of the figures 1 and 2 is low - I suggest improving it.
Overall, the results are presented clearly.
4. Materials:
Glutamate assay kit - please add CVs and detection limit.
5. Discussion - please summarize the limitations of your own research. What is the clinical impact of your own observations?
In general, a very interesting article, well written and suggesting that this topic must be addressed in the future research.
Author Response
- Title - I suggest deleting in vitro and in vivo.
Response 1: Thanks for the reviewer’s suggestions, we already deleted “in vitro and in vivo” in title.
- Introduction:
I suggest adding literature to the first sentence e.g. Ann Clin Biochem. 2017;54(3):378-385.
STAT1 - please provide full name (signal transducer and activator of transcription 1).
Give a name "Emodin" with a lowercase.
Response 2: We have added this reference on page 2, line 55.
- Results:
the quality of the figures 1 and 2 is low - I suggest improving it.
Overall, the results are presented clearly.
Response 3: Figures 1 and 2 are revised on page 4&5.
- Materials:
Glutamate assay kit - please add CVs and detection limit.
Response 4: The CV and detection limit are added on line 336-337.
- Discussion - please summarize the limitations of your own research. What is the clinical impact of your own observations?
Response 5: Limitations of this study is summarized on page11, line217-224. Clinical impact is described on page12, line 231-247.
Reviewer 2 Report
The author proposed that "Neuroprotective effects of Emodin against ischemia/reperfusion injury through the ERK-1/2 signaling pathway in vitro and in vivo". However, there is only one Western blot data in Figure 2C and 2D showed that Emodin increases p-ERK-1/2. The experiments should be performed to test the attenuated beneficial effect of Emodin after inhibition of ERK-1/2 through pharmacological or genetic methods. Therefore, the evidence is largely lack to support the conclusion in the manuscript that Emodin protects against ischemia/reperfusion injury through ERK-1/2 signaling.
Major concerns:
- In Figure 1, it seems that the hypoxia model was used according to the graph, but figure legend showed that OGD/RP model was used. Obviously, Hypoxia is not OGD/RP or OGD. The author should make it clear and consistent throughout the manuscript.
- According to the data showed in Figure 1B, it seems that 3μM of Emodin has no neurotoxicity effect, but a lower dose (1μM) and a higher dose (10μM) of Emodin significantly induced neurotoxicity of PC12 cells. This data is very weird. What are the possible reasons?
- The description in the manuscript is not consistent with the results showed in Figures. For example, Figure 2D, but not Figure 1D showed the inhibitory effect of emodin on ROS. Figure 2C and 2D, but not Figure Figure 2A and 2B showed the effect of emodin on ERK, GLT-1, etc.
- There is no scale bar in Figure 3A and 3B, and Figure 4A. The image quality in Figure 3C and 4A are too weak. The author should replace them with clearer images.
- It is commonly believed that the glutamate transporter GLT-1 is highly expressed in astrocytes but also in neurons, and the extracellular glutamate is mainly uptake by astrocytes. Therefore, it makes more sense to detect the effect of Emodin on GLT-1 in astrocyte, but not PC12 neuronal cells. The author should detect GLT-1 expression in different brain cells include neurons, astrocytes, microglia by immunostaining.
- It had been proposed in Figure 2 that “the effects of Emodin treatment of PC12 cells after OGD/RP on ROS production, glutamate release, and apoptosis are regulated through the ERK-1/2 signaling pathway”. However, there is no evidence to show the role of ERK-1/2 signaling in regulating ROS production, glutamate release, and apoptosis in the study.
Author Response
Major concerns:
In Figure 1, it seems that the hypoxia model was used according to the graph, but figure legend showed that OGD/RP model was used. Obviously, Hypoxia is not OGD/RP or OGD. The author should make it clear and consistent throughout the manuscript.
Response: Thanks for the reviewer’s comments. We replaced OGD/RP with OGD/Hypoxia in figure 1 (page 4).
According to the data showed in Figure 1B, it seems that 3μM of Emodin has no neurotoxicity effect, but a lower dose (1μM) and a higher dose (10μM) of Emodin significantly induced neurotoxicity of PC12 cells. This data is very weird. What are the possible reasons?
Response: We apologize for not carefully presenting data in figure 1B. Both 1 & 3 μM of Emodin have no neurotoxicity effect, we have revised figure 1B on page 4.
The description in the manuscript is not consistent with the results showed in Figures. For example, Figure 2D, but not Figure 1D showed the inhibitory effect of emodin on ROS. Figure 2C and 2D, but not Figure Figure 2A and 2B showed the effect of emodin on ERK, GLT-1, etc.
Response: We have corrected the results on page 3 line 103, page 4 line 118, 120, 125.
There is no scale bar in Figure 3A and 3B, and Figure 4A. The image quality in Figure 3C and 4A are too weak. The author should replace them with clearer images.
Response: We have added scale bar in Figure 3A, 3B, and Figure 4A, and replaced with clear images of Figure 3C and 4A on page 9 and 10.
It is commonly believed that the glutamate transporter GLT-1 is highly expressed in astrocytes but also in neurons, and the extracellular glutamate is mainly uptake by astrocytes. Therefore, it makes more sense to detect the effect of Emodin on GLT-1 in astrocyte, but not PC12 neuronal cells. The author should detect GLT-1 expression in different brain cells include neurons, astrocytes, microglia by immunostaining.
It had been proposed in Figure 2 that “the effects of Emodin treatment of PC12 cells after OGD/RP on ROS production, glutamate release, and apoptosis are regulated through the ERK-1/2 signaling pathway”. However, there is no evidence to show the role of ERK-1/2 signaling in regulating ROS production, glutamate release, and apoptosis in the study.
Response: We agree with reviewer’s comments that GLT-1 is highly expressed in astrocytes and it’s important to explore the effect of Emodin on astrocytes. However, in this study we focus on the neuroprotective role of Emodin and found that Emodin indeed decreased the neural death after ischemia through regulation of ROS production and glutamate release. The detailed molecular mechanisms of Emodin still unclear, therefore, we intend in the future to perform experiments to understand the role of Emodin in astrocytes and mechanisms underlining the decreased neural death via Emodin treatment.
Reviewer 3 Report
This study examines mechanisms by which Emodin, an anthraquinone derivative, regulates neuronal death associated with focal brain ischemia in vitro and in vivo. The present study builds on the previous studies by a few other groups which showed that Emodin protects neurons against cerebral ischemia-reperfusion injury by reducing glutamate toxicity. The present study represents an advance over the previous papers in that it suggests an increase in expression of glutamate transporter GLT-1 mediated by Emodin as a key factor that reduces extracellular glutamate which in turn, decreases ROS generation and protects neuronal death. Here the authors report that Emodin reduces ROS generation and extracellular glutamate in PC12 cells subjected to OGD, an in vitro model of ischemic stroke. The results further show that OGD reduces p-ERK-1/2, GLT-1 and Bcl-2 while increases activated form of caspase-3. Emodin restores the protein level of p-ERK-1/2, GLT-1, Bcl-2 and activated form of caspase-3 almost by control level or a little less. Importantly, in vivo results show administration of Emodin attenuates infarction sizes after MCAO in rats and improves the recovery in body asymmetry. To prove the molecular mechanism showing in vitro could be relevant to an in vivo animal model, an increase in GLT-1 protein level was shown in rat brain sections and images from Western blot analysis. The authors conclude that neuroprotective effect of Emodin could possibly be occurred by increasing GLT-1 level which is known to be regulated by activation of ERK-1/2.
This is an interesting study which adds an evidence of the neuroprotective effects of Emodin and sheds new light on how Emodin protects neuronal death associated with focal brain ischemic injury.
There are, however, several issues that should be addressed by the authors:
Major points:
- GLT-1 is mainly expressed in glial cells although found low levels in the axon-terminals of neurons. It is important to examine whether Emodin induces GLT-1 protein level in glial cells (astrocytes) or in neurons. The Figure 4 showed that Emodin induced GLT-1 protein level but it is unclear if it is in glial GLT-1 or neuronal GLT-1. The high magnification of images or double labeling of GLT-1 with neuronal marker or astrocyte marker would be possible approaches to address this issue. Along with this concept, the model that authors suggested in Figure 5 is also unclear that the cartoon of cell is a glia or neuron, if it is glia, cross talk between glia and neuron has to be illustrated, if it is neuron, it should be separately illustrated with presynaptic and postsynaptic neurons. It doesn’t make sense to show GLT-1 is increased in the membrane and intracellular glutamate level is decreased in one cell.
In addition, the subcellular location of GLT-1 as well as its function should be described or discussed in the text.
- The authors suggested the ERK-1/2 signaling pathway including GLT-1 activation as a key mechanism of the protective effects of Emodin in vitro and in vivo, however, only data shown in vivo (Figure 4) is GLT-1 protein level. It would be nice to include data for ERK1/2, Bcl-2 and caspase-3 as the authors nicely showed them in PC12 cells with the bar graphs.
- In Figure 2C (could be Figure 2A), p-ERK1/2 should be normalized to total ERK. (then, if preferred, to β-actin again).
Minor points:
- In page 2, line 72, at the end of the sentence, citation needs to be added.
- In page 2, line 78, at the end of the sentence, “….MCAO) models.”, citation needs to be added.
- In page 3, lines 103-104, the sentence “After Emodin treatment, the ratios of ERK-1/2 and GLT-1 expression were greater than 1 for both the normoxia and hypoxia group” is incorrect; the protein expression of GLT-1 in the Emodin treated hypoxia group is not greater than 1.
- In Figure 2, the figures should be rearranged to be matched with manuscript.
- Figure 2A should be Figure 1D.
- Figure 2B should be Figure 2C.
- Figure 2C and 2D should be Figure 2A and 2B.
- Figure legends should be changed accordingly.
- In Figure 2 legends, description about #P and ##P needs to be added.
- In Figure 3 legends, description of how many animals were used in each group and experiment needs to be added.
- In Figure 4B, it looks there are two bands for GLT-1 and pointed lower band. It should be double checked if this is correct. Also, since this is representative image, a bar graph needs to be added.
- In page 7, line 221, 15 mg/Kg should be fixed to 15 mg/kg.
- In page 9, line 272, 200 mL/well should be fixed to 200 µl/well.
- In page 9, line 286, 50 mg should be fixed to 50 µg.
- In page 9, line 290, ‘TBST-Tween 20’ should be fixed either ‘TBST’ or ‘TBS-Tween 20’.
- Antibody information should be added in the Materials and Methods.
Author Response
Major points:
GLT-1 is mainly expressed in glial cells although found low levels in the axon-terminals of neurons. It is important to examine whether Emodin induces GLT-1 protein level in glial cells (astrocytes) or in neurons. The Figure 4 showed that Emodin induced GLT-1 protein level but it is unclear if it is in glial GLT-1 or neuronal GLT-1. The high magnification of images or double labeling of GLT-1 with neuronal marker or astrocyte marker would be possible approaches to address this issue. Along with this concept, the model that authors suggested in Figure 5 is also unclear that the cartoon of cell is a glia or neuron, if it is glia, cross talk between glia and neuron has to be illustrated, if it is neuron, it should be separately illustrated with presynaptic and postsynaptic neurons. It doesn’t make sense to show GLT-1 is increased in the membrane and intracellular glutamate level is decreased in one cell.
In addition, the subcellular location of GLT-1 as well as its function should be described or discussed in the text.
Response: Thanks for the reviewer’s comments. We agree with reviewer’s points that it is important to examine the effect of Emodin on glial cells. We intend in the future to perform animal experiments to understand the role of Emodin in glial cells and mechanisms underlining the decreased neural death via Emodin treatment. As for figure 5, we have revised the cartoon according to reviewer’s comments.
In the model of ischemic brain disease and heart disease, leading to overproduction of ROS and creating oxidative stress [1-3], following damages DNA, proteins, and membrane lipids then cause cell death. The natural phenols such as tyrosol or Emodin have antioxidation capability that protected cells against injury due to oxidation in vitro [4,5]. Previous review in 2015 has shown that tyrosol reduced of ROS accumulation and involved in ERK and Bcl-2 signaling pathway, therefore we suggest Emodin have similar protective effects and possible mechanism in the MCAO model.
- Yang JL, Mukda S, Chen SD Diverse roles of mitochondria in ischemic stroke. Redox Biol. 2018, 16, 263-275.
- Halliwell, B. Reactive oxygen species and the central nervous system. J. Neurochem. 1992, 59, 1609–1623.
- Ambrosio, G.; Tritto, I. Reperfusion injury: Experimental evidence and clinical implications. Am. Heart J. 1999, 138, S69–S75
- Sun L, Fan H, Yang L, Shi L, Liu Y. Tyrosol prevents ischemia/reperfusion-induced cardiac injury in H9c2 cells: involvement of ROS, Hsp70, JNK and ERK, and apoptosis. Molecules. 2015, 20, 3758-3775.
- Chang R, Zhou R, Qi X, Wang J, Wu F, Yang W, Zhang W, Sun T, Li Y, Yu J. Protective effects of aloin on oxygen and glucose deprivation-induced injury in PC12 cells. Brain Res Bull. 2016, 121, 75-83.
The authors suggested the ERK-1/2 signaling pathway including GLT-1 activation as a key mechanism of the protective effects of Emodin in vitro and in vivo, however, only data shown in vivo (Figure 4) is GLT-1 protein level. It would be nice to include data for ERK1/2, Bcl-2 and caspase-3 as the authors nicely showed them in PC12 cells with the bar graphs.
Response: We have added the expression of ERK1/2, Bcl-2 and caspase-3 in figure 4B (page10).
In Figure 2C (could be Figure 2A), p-ERK1/2 should be normalized to total ERK. (then, if preferred, to β-actin again).
Response: According to reviewer’s comment, we have added the expression of total ERK in Figure 2B and normalized the p-ERK with total ERK and b-actin in Figure 2C. (page 5)
Minor points:
In page 2, line 72, at the end of the sentence, citation needs to be added.
Already added references 18, 19 (line 85).
In page 2, line 78, at the end of the sentence, “….MCAO) models.”, citation needs to be added.
Already added references 23 (line 85).
In page 3, lines 103-104, the sentence “After Emodin treatment, the ratios of ERK-1/2 and GLT-1 expression were greater than 1 for both the normoxia and hypoxia group” is incorrect; the protein expression of GLT-1 in the Emodin treated hypoxia group is not greater than 1.
Already corrected (line 120).
In Figure 2, the figures should be rearranged to be matched with manuscript.
Figure 2A should be Figure 1D.
Figure 2B should be Figure 2C.
Figure 2C and 2D should be Figure 2A and 2B.
Figure legends should be changed accordingly.
In Figure 2 legends, description about #P and ##P needs to be added.
Already corrected in Figure 2 on page 6.
In Figure 3 legends, description of how many animals were used in each group and experiment needs to be added.
Already add animal numbers in Figure 3 legends on page 9.
In Figure 4B, it looks there are two bands for GLT-1 and pointed lower band. It should be double checked if this is correct. Also, since this is representative image, a bar graph needs to be added.
Already added scale bar in Figure 4A and replaced WB data in Figure 4B.
In page 7, line 221, 15 mg/Kg should be fixed to 15 mg/kg.
Already corrected (page 12, line 255)
In page 9, line 272, 200 mL/well should be fixed to 200 µl/well.
Already corrected (page 13, line 306)
In page 9, line 286, 50 mg should be fixed to 50 µg.
Already corrected (page 13, line 320)
In page 9, line 290, ‘TBST-Tween 20’ should be fixed either ‘TBST’ or ‘TBS-Tween 20’.
Antibody information should be added in the Materials and Methods
Already corrected (page 13, line 324)
Round 2
Reviewer 2 Report
In this revision, although the authors addressed some of my concerns, I still have concerns about the conclusion that “Emodin increased Bcl-2 and GLT-l expression, but suppressed activated-caspase 3 levels through the ERK-1/2 signaling pathway”. The evidence from the study showed that Emodin can increase ERK-1/2, concomitantly increase Bcl-2, GLT-1, and reduce active caspase-3, but there is no direct evidence showed ERK-1/2 signaling is involved in Emodin induced Bcl-2 and GLT-1. The study can only conclude “Emodin increased Bcl-2 and GLT-l expression, but suppressed activated-caspase 3 levels may be through activating ERK-1/2 signaling pathway”
Author Response
Response: Thanks for reviewer’s suggestion, we have revised the conclusion “Emodin has neuroprotective effects against ischemia/reperfusion injury both in vitro and in vivo may be through activating ERK-1/2 signaling pathway.
” in abstract (line42-43).
Reviewer 3 Report
Major points:
- It is understandable that authors intend to examine the role of Emodin in glial cells and mechanisms by which Emodin decreases neural death in the future. In this case, the authors could not rule out that GLT-1 in glial cells could uptake extra cellular glutamate which possibly in turn, protect neurons against excitotoxicity. One possible model for Figure 5 is suggested below, it is not perfect but authors could modify and include some ideas from the suggested model. Figure 5 also needs figure legend.
- The bar graph needs to be added in Figure 4B.
All minor points are well addressed.

Author Response
Response: Thanks for reviewer’s comments, we have revised figure 5 and add figure legend according to reviewer’s suggestion. Also, we have added bar graph in Figure 4C.